# Physical and mental health professionals perspectives of providing mental health care for young people: A qualitative interview study

Jessica Folwell[1], Dihini Pilimatalawwe[1], Julia Mannes[1], Sara O'Curry[2,3], Cathy Walsh[2,3], Jenny Gibbs[1], Debbie Critoph[1], Isabella Morse[1], Robbie Duschinsky[1], Tessa Morgan[1]*

1 Department of Public Health and Primary Care, Child Health and Development Group, The University of Cambridge, West Forvie Site, Cambridge, United Kingdom, 2 Cambridge University Hospital, Cambridge, United Kingdom, 3 Cambridge and Peterborough NHS Foundation Trust, Cambridge, United Kingdom

* tlm32@medschl.cam.ac.uk

## Abstract

Rates of poor mental health among children and young people are rising globally, with physical health professionals increasingly expected to respond to psychiatric needs. Despite this shift, limited research has explored how these professionals experience and manage mental health presentations, particularly in paediatric settings. This study examines the challenges and opportunities faced by staff supporting young people with mental health needs on paediatric hospital wards, within a system that often treats physical and mental health separately. We conducted a secondary analysis of 31 one-off semi-structured interviews, conducted with 16 mental health and 15 physical health professionals. Using reflexive thematic analysis, themes were iteratively refined in dialogue with NHS collaborators, a senior qualitative researcher, and interview participants to ensure analytic rigour and relevance. Professionals reported a widening gap between the complexity of young patients' mental health needs and the limited expertise available on physical health wards. Three themes emerged: (1) "We all feel out of our depth," reflecting feelings of being underprepared and overwhelmed; (2) "A mental health waiting room," highlighting wards being used as temporary spaces while patients await psychological care; and (3) "We're the place to keep them safe," revealing a primary focus on immediate risk management. Physical health professionals reported feeling unprepared to support young patients with mental health needs, often managing self-harm, suicidality, and eating disorders without specialist training. Both physical and mental health professionals emphasized a need for trauma-informed, non-stigmatizing communication and emotional support for staff. Barriers to integrated care within these two trusts included digital system incompatibility, understaffing, and limited psychiatric liaison. Findings highlight the urgency of cross-disciplinary training, supervision, and structural investment to support compassionate, coordinated care for young people with complex mental and physical health needs.

**Data availability statement:** The participants of this study did not give written consent for their data to be shared publicly, so due to the sensitive nature of the research supporting data is not available. The full interview guide has been uploaded alongside the requirements for resubmission as a supplementary file. The codebook has been included in the manuscript, in the form of the theme map seen in Figure 2. An anonymised quote bank, including the additional quotes we have added as part of this resubmission, can be accessed by email request to the corresponding author or to the departmental data manager James Brimicombe: jb16@medschl.cam.ac.uk.

**Funding:** This work was funded by the Addenbrooke's Charitable Trust, Grant Number: 900345 (RD, CW. SC). The funders had no role in study design, data collection and analysis, decision to publish, or preparation of the manuscript. https://act4addenbrookes.org.uk/.

**Competing interests:** The authors have declared that no competing interests exist.

## Introduction

The growing rates of poor mental health among children and young people under 18 have been recognized as a global public health problem [1,2]. In England, an estimated one in five young people meet the diagnostic criteria for a mental health disorder [3], a figure that has risen significantly since the COVID-19 pandemic [4,5]. Mental health needs of young people have become more complex, becoming intertwined with wider sociodemographic determinants such as poverty or facing stigmatisation [6,7]. Attributed in part to long waiting lists for mental health services, physical healthcare professionals are increasingly attending to a range of young people's psychiatric needs [8]. Emergency departments worldwide report increased young patient admission due to mental health needs [9,10], with mental health related presentations in England rising by 200% between 2010–2020 [11]. Some common types of hospitalisations can require both physical and mental health attendance, such as in the cases of suicide attempts, self-harm or eating disorders [12]. The mounting recognition of mental health needs being complex and not neatly fitting into the current siloed healthcare system [13,14], has led some to call for integrated care systems to grapple with the growing complexity of need [15,16].

At present there is little known about physical health professionals' experiences of responding to the needs of mental health patients admitted to hospital wards [17]. There are even fewer studies about those who attend to children and adolescents [18]. This gap is particularly concerning as physical health staff are increasingly on the frontline of mental health provision [9], despite specialised training in addressing mental health concerns only becoming a mandatory component of the UK's foundation programme (curriculum for doctors in their first two years after qualification) in 2021 [19]. Of the scant evidence available, studies indicate that physical health professionals perceive that they lack the necessary skills to handle and support mental health related needs that appear on their caseloads [20]. Many have called for clear training on how professionals should operate within integrated care settings [21,22]. Health professionals in general often reported feeling deskilled when required to manage mental health conditions, often citing a lack of specialized training in psychiatry, diagnostic uncertainty, or reduced confidence in their ability to provide appropriate care [23]. Some staff report role confusion, as it was unclear what their responsibilities as physical healthcare professionals were when patient needs were not solely physical in nature [24].

Additionally, there are concerns over whether mental and physical health needs are treated with parity within such systems [25]. While efforts have been made to promote equal prioritization, mental health services often face longer wait times, lower funding and lower resource allocation [26]. Evidence from health care professionals and young people themselves also indicate that mental health care remains stigmatised [27,28]. In Radez and colleagues' [29] systematic review of 53 studies 92% of young people identified perceived social stigma and embarrassment as significant barriers to seeking professional mental health care. Given this lack of parity in service provision and the continued stigma surrounding mental health, it is perhaps unsurprising that young people are turning to physical health care systems to attend to mental health needs [30,31].

In this context, there is growing enthusiasm for more holistic, integrated approaches to young people's mental health, moving beyond narrow medical models, that recognise the interdependence and interplay of biological, psychological, and social factors [15]. Building on this growing evidence base, this study examines the opportunities and challenges faced by healthcare professionals supporting mental health needs of children/young people in paediatric wards.

## Methodology

### Study design

This research was commissioned by two NHS trusts in preparation for the development of a new Children's Hospital in England. The research itself was conducted by academic researchers independent from the NHS trusts. One experienced female post-doctoral qualitative researcher (TM) who was a non-clinician conducted all interviews. TM was not affiliated with either NHS trust. This was outlined on the participant information sheet and explicitly restated at the beginning of each interview. Two members of the research team were affiliated with both trusts (SO, CW). To protect participant confidentiality, only anonymised data were shared with the wider research team.

### Ethics statement

This study received ethics approval, including permission to conduct this secondary analysis, from the Cambridge Psychology Research Ethics Committee (Application No: PRE.2022.118). Data collection was conducted between November 2023-March 2024. All participants received a copy of the participant information sheet and had opportunities to ask questions about the study before taking part. All participants signed a written consent form before they took part in the research which included explicit permission for data accessibility for students working with the Child and Health Development research group.

### Data collection

Recruitment was supported by both NHS trusts. Clinical leads forwarded an email invitation along with the interview questions to inform staff about the study. Participants were invited to take part in a study about their experiences and expectations of integrated working between physical and mental health trusts. Interested staff were required to email the independent researcher to express their interest. To protect participants confidentiality, no information about who took part was shared with any clinical leads. By taking part, professionals went into a draw for a £300 voucher.

One-off semi-structured interviews were conducted between November 2023-March 2024. A total of 31 participants took part including 16 mental health professionals from one trust and 15 physical health professionals from another (See Table 1). The mental health professionals interviewed worked within Child and Adolescent Mental Health Services (CAMHS), the UK's National Health Service (NHS) specialist mental health teams for children and young people (up to 18). CAMHS provides free assessments, therapy, medication, and support for complex emotional/behavioural issues through a multidisciplinary team comprising of psychiatrists, psychologists, nurses and occasionally social workers [32].

Participants were purposively sampled to ensure equal representation across each trust and a diversity of professional backgrounds was reflected. Participants included allied health professionals, nurses, team managers, paediatricians, psychiatrists and psychologists.

All participants were asked the following questions, which were sent ahead of time to support the transparency of the project. This sub-analysis focused principally on question one but drew in where relevant responses from question three (See S1 File for full interview guide):

1) In your current role, do you interact with [name of the other trust] staff?

 1a) What is your experience of those interactions?

3) How would you hope to interact/work together in the Children's Hospital?

**Table 1. Sample characteristics.**

| Gender | n |
|---|---|
| Female | 31 |
| Physical Health Roles | |
| Paediatricians | 3 |
| Nurse and Health care assistants | 5 |
| Managers | 2 |
| Allied Health Care Professionals | 5 |
| Mental Health Roles | |
| Nurse and Health care assistants | 7 |
| Psychologists | 3 |
| Psychiatrist | 4 |
| Managers | 1 |
| Community CAMHS | 1 |

All interviews were audio-recorded and ranged from 10 minutes to one hour. Audio was transcribed verbatim using the online software TRINT. All transcriptions were reviewed and cleaned by the interviewer. One transcript was returned to a participant on request for them to edit sensitive, non-study related information. All transcripts were anonymised by the interviewer. No identifiable information was shared beyond the interviewer, including to the wider research team.

## Data analysis

The present study conducts a secondary analysis of the 31 interview transcripts from physical healthcare, mental healthcare or allied health professionals from two NHS trusts (see Table 1). No staff identified themselves as dually trained in physical and mental health. Staff had the option of being interviewed individually or in a group. All three group interviews took place at the group's place of work. The allied physical healthcare staff and all mental health staff requested group interviews as this was convenient due to the limited availability of staff during the working day to partake in an interview. Individual interviews have the benefit of building enhanced rapport between the interviewee and interviewer thereby creating an open space for participants to express their views [33]. However, group interviews, whilst not providing the same level of depth, enable participants to discuss sensitive topics in a collective manner, making the process feel safer and enabling individuals to consider more structural components in their narratives [34]. Eight interviews were conducted in person including one group interview of four staff. A further seven individual interviews were conducted over zoom at a time of the participants choosing. The sample size of 31 was deemed sufficient to obtain information power, ensuring relevant, rich, and specific data that aligns with the present study aims [35].

Several interviews were necessarily shorter in time due to the pragmatic realities of professionals only being available for interviews during their short lunch breaks. The shorter interviews were primarily conducted with student nurses and healthcare assistants whose more limited experience on the ward had resulted in shorter conversations on their experience. Interviews were nonetheless information rich as the interview schedule was concise yet focused, enabling participants time to engage meaningfully with core topics despite time restrictions. Offering zoom interviews after work hours allowed other staff to explore their perspective through longer interviews, which helped to the depth of the data.

We used reflexive thematic analysis which is an inductive and flexible qualitative approach (See Fig 1 for Analysis Steps) that identifies "patterns of shared meaning, cohering around a central concept" [36]. This method emphasizes the researcher's active role in knowledge production [37]. According to Braun and Clarke [37], reflexivity occurs at the intersection of (1) the dataset, (2) the theoretical assumptions of the analysis, and (3) the analytical skills and resources of the researcher. We reflect on each of these below.

PLOS Mental Health

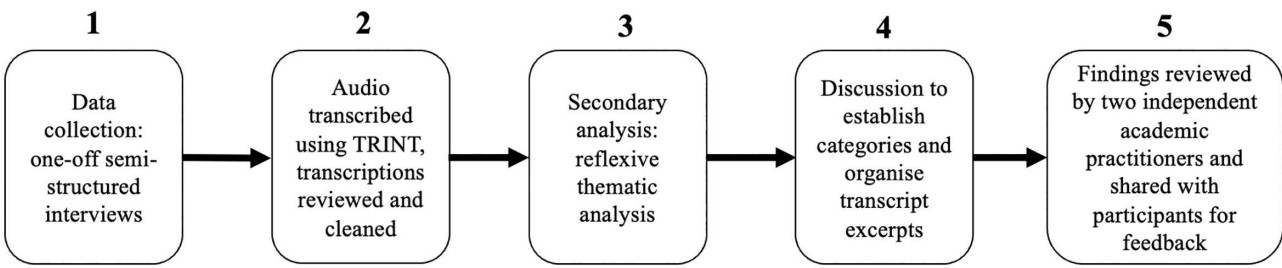

**Fig 1. Analysis steps.**

Once data collection and transcript anonymisation were complete, two female researchers (TM the interviewer and JF a fourth-year medical student) familiarised themselves with the data by separately reading through and open coding each of the transcripts. Codes were understood as 'central organising concepts' that reflected the researcher's interpretation of patterns of meaning across the dataset [36]. This process supported our reflexive and thoughtful engagement with the analytic process as it allowed us to ask questions of each other's interpretations which helped to clarify meanings.. This process involved the lead author, JF a fourth-year medical student, reflecting on her clinical training and observations from her placements. Initial ideas were then shared with a senior male qualitative researcher (RD) to refine and carefully cluster emerging concepts into provisional themes. Next, broad discussion categories frequently mentioned by staff (e.g., education and skills, patient experience) were established, and relevant transcript excerpts were organized within these categories. The researchers TM and JF, alongside a female research assistant specialising in intersectional qualitative methodologies (DP), then explored the latent themes emerging from each category, initially focusing on how they addressed physical and mental health staff's perceptions of patients' needs, professionals' responsibilities and barriers to patient care. Outliers were discussed as a research team, with negative cases also presented in the paper [38,39]. Our analytic stance allowed for deeper insights to be drawn from even brief interviews, as it focuses on the significance and implications of what is said rather than the quantity of data produced. Top-level findings were shared at a series of six consultation meetings facilitated by the interviewer and collaborators from the NHS trust to ensure their trustworthiness.

To support the rigour and relevance of the findings and recommendations two female independent academic practitioners, JG a children's social worker and DC a paediatric nurse, triangulated the interpretation of findings from JF, DP and TM to help reduce potential bias. To further support the transparency and credibility of findings, the manuscript was shared with participants who had the opportunity to provide feedback (two did so). Findings benefited from rounds of feedback at three stakeholder meetings with professionals from both trusts. To ensure participant confidentiality quotations have been anonymised and are attributed to which interview session they took part in.

## Results

Physical health professionals strongly expressed that mental health expertise was lacking in paediatric physical health wards, despite a rising level of mental health-related need among their patients. The key theme clusters and their corresponding sub-themes are presented in Fig 2. The first theme identified was *"We all feel out of our depth"*, captures professionals' sense of being unprepared and sometimes overwhelmed when managing young people's mental health presentations. The second theme *"a mental health waiting room"* reflected a perception that physical health wards have become placeholders for young people with mental health needs to wait for psychological support. The third and final theme *"We're the place to keep them safe"*, which indicates the principal obligation many health professionals felt towards mental health patients.

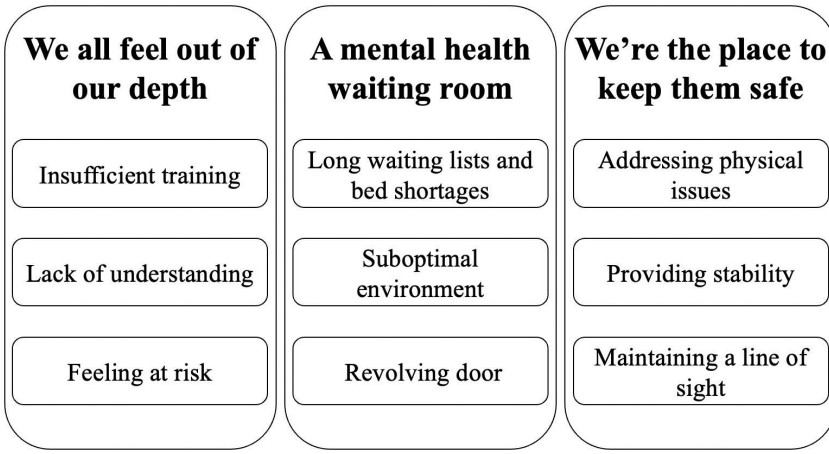

**Fig 2. Theme map.**

### "We all feel out of our depth"

Professionals frequently expressed concerns about inadequate training in communicating with mental health patients, particularly those with eating disorders and those in crisis. Physical health professionals were afraid that they were "going to slip up in the chaos of the day and say something that's really going to give them (patients) a step back" (PH12). Many felt unprepared, unsure of "what to say [and] what not to say" (PH10), often relying on informal guidance from colleagues, such as dietitians advising, "don't mention this at mealtime [or] say it this way" (PH10). Mental health professionals noted how they felt physical health professionals fear of miscommunicating with patients with self-harm related needs came from them not having sufficient knowledge of how to respond to these types of young people (MH2). Owing to any substantive formal training, these patient interactions often left physical health professionals feeling deskilled, stressed, and questioning how to "do the best job that [they] could" (PH9). Many physical health staff expressed concerns that "it doesn't feel like we're doing a great deal for them" (PH6).

Multiple physical health professionals shared concerns around not feeling confident in their understanding of the underlying reason for more severe mental health presentations. Some professionals perceived mental health concerns as primarily a mechanism of communication and help-seeking, as young people often presented to the emergency department "in crisis" (PH6, PH12) as a way to get "someone … to listen to them" (PH8). One physical health professional noted how some other physical health staff members held biased views towards mental health needs as "so many people brush mental health off…they're actually fine, just attention seeking" (PH8). A mental health professional shared similar disappointment in the "common stigma" displayed by their peers as they "thought it would be less present" in their field (MH1). However, this was not the consensus as in a minority of our interviews physical health professionals expressed more negatively skewed descriptions of mental health needs. Often times these physical health professionals discussed how the "competitive nature of the eating disorder patients" (PH1) or the presence of suicide "trends" among individuals from the "same friendship groups" (PH10) made supporting mental health needs particularly difficult. This perceived desire to be "the most sickest person" (MH2) amongst young people with mental health needs was also discussed by a mental health professional. There was lesser acknowledgement about other functions of such behaviour such as negative emotion regulation or self-punishment.

Physical health professionals felt that caring for "particularly unwell...challenging" mental health patients put them at risk and potentially resulted in "quite volatile, quite dangerous situations" (PH12). Multiple professionals discussed difficulties attending to patients when "the child themself is not complying or they're being disruptive" (PH10). A few professionals also discussed the strain of balancing different patients' needs on under-staffed shifts:

"If you do have a really challenging [patient]…who attempts to abscond and stuff like that, then you're taking staff members away from…like the children on the ward who are here for surgery" (PH8).

These instances left some physical health professionals feeling that they "didn't sign up for this," given how "emotionally draining" it was to provide such care (PH12). A few professionals, both mental and physical health, also shared concerns about the impact on other "families and children on the wards" for whom these incidents could be "scary" and "stressful" (PH3).

**"A mental health waiting room"**

Both physical and mental health professionals expressed a shared concern that paediatric wards increasingly served as waiting room for CAMHS assessment and inpatient care. Multiple physical health professionals viewed this as a result of long waiting lists and widespread "shortages all around" (PH6) of Tier 4 mental health inpatient beds, the most specialized level of care available within CAMHS typically used by young people with severe, complex, and persistent mental health problems like psychosis [40]. This has resulted, as one physical health nurse phrased it, in patients having to "chillax, in a bed, waiting for [mental health services] to arrive" (PH12) before any decisions could be made. Such delays between services were attributed to patients who "[didn't] have a medical need" having to be admitted overnight because "they haven't been able to access appropriate assessment" that day (PH3).

One paediatrician shared positive experiences of working alongside a liaison psychiatry team which underpinned her view that mental and physical assessments should be done "side by side …which should result in a shorter and more efficient admission time" (PH3).

Some of these challenges stemmed from logistical constraints. Physical health professionals noted how misalignment of team working hours resulted in young people having to wait "until the next morning" as the crisis team "only takes referrals up to 930pm" (PH3). Communication challenges deriving from separate computer systems between services were described by all as a "massive issue" (PH6) preventing professionals from accessing relevant information such as a patient's CAMHS assessments. Mental health professionals also acknowledged how the current system "is quite patchy" (MH5) and also expressed frustration at the "waiting period" (MH2) they faced when trying to access physical healthcare for their patients due to being "treated like an individual who just goes to A&E" (MH2).

Some physical health professionals felt that the physical health ward was overall not a suitable space of care for mental health patients, as they "just don't always think it's the right environment" (PH1). Some mental health professionals concurred that they felt that the physical health wards were not equipped to handle the safety concerns of young people with mental health needs (MH6). Other physical health professionals noted that young people presented with complex needs that fell out of the remit of physical or mental health and were more appropriate for social care (PH6). A few professionals acknowledged how the hospital may be "the only option" when appropriate social care was not in place and "everything else [is] so much more unhappy or much more unsafe for [the young people]" (PH12). A few physical health professionals expressed frustration that no system could adequately support these young people, turning the ward into a "revolving door" (PH12) for a certain group of young people with complex needs repeatedly returned "in either a more dire situation or a more unwell situation, or [facing] prolonged, repeated admissions" (PH12, PH10).

**"We're the place to keep them safe"**

All participants noted the growing presence of young people's mental health needs on physical health wards. One participant concisely summarised that "mental health is creeping into physical health so much more and there are times when, you know, the majority of our patients are mental health patients" (PH1). Professionals noted increased presentations of patients with eating disorders, patients who had self-harmed and those who had attempted suicide by overdose.

Professionals perceived their role was to address the physical issues that arose as a consequence of mental health concerns. Tasks involved providing medication to counteract overdoses, medical interventions following self-harm or nutritional support in the cases of eating disorders (PH10). In addition, they felt they offered young people a "place of safety" (PH12) and "stability" (PH6). A few participants recounted how patients had shared this sentiment as "[the patients] were actually saying to the nurses or us like, I want to stay here, I don't feel safe going home. I don't trust myself if I go home" (PH8). Physical health professionals felt responsible for patients' safety, as they "are the ones that are physically there with those patients, so [they] have to try [their] best" (PH12). In addition, physical health professionals viewed it as their duty to get their patients "medically fit" (PH1) as this was perceived as a requirement from mental health professionals to begin their assessments. Mental health professionals confirmed that, although there was often a long waiting time to receive physical health support, such support was provided well by physical health professionals. One psychologist gave the example of when they "send a patient [to a physical health ward] because they were not eating" (MH2), the patient came back with improved blood glucose levels, suggesting the physical health staff were able to successfully get the young person to eat something.

Both mental and physical health professionals viewed patient safety on the ward as one of their principal responsibilities. Some physical health professionals talked about always having to maintain a "line of sight" on patients (PH12) typically by putting them in the bay right in front of the nurses' station (PH10). Physical health staff also talked about formalised observation called "specialing" where a staff member, closely observes those assessed as high risk due to potentially significant cognitive impairment, challenging behaviour, risk of falls, risk of self-harm, or risk to others [41]. Physical health professionals noted how during "specialing" they, mostly healthcare assistants, have to sit "with [patients] all day and all night" (PH8). Mental health professionals described an identical level of monitoring with their own patients (MH2, MH3, MH6). This constant surveillance was considered "emotionally draining" but necessary to ensure patient's safety (PH8).

## Discussion

There has been growing academic and policy attention to the mental health challenges facing young people which have been connected to rising rates of mental health admissions in physical health settings [9,11]. This study builds on the limited literature to date around physical health professionals' experiences of caring for young people with mental health needs in traditionally physical health spaces, such as emergency departments and paediatric wards [18,17]. Our study only captures a snapshot of a healthcare system in transition, as changes have occurred since the data was initially collected. Influenced in part by early findings shared with the NHS Trusts, procedural and logistical changes have occurred. Resultantly, the uncertainties and anxieties reported by participants may have evolved or been mitigated through new policies, training, or structural adjustments. However, while specific challenges may have shifted, the broader themes regarding the preparedness of physical health professionals to support young people with mental health needs remain highly relevant.

Our principal finding was that physical health professionals felt inadequately prepared and supported when attending to the complexity of mental health patients' needs who made it onto their wards. Our analysis suggests that professionals primarily saw their role as one of observation, stabilization, and ensuring safety. Building on previous findings [12], professionals reported that the primary presentations among their patients were self-harm, suicide-related behaviours, and eating disorders. Chief areas of concern for physical health professionals reported by both physical and mental health staff included ways of supporting mealtime for patients with eating disorders and ways of sensitively caring for patients in crisis. These findings align with previous studies of healthcare professionals who experienced a sense of limited capability when required to manage concerns beyond their specialized training [42,20], especially when situations are particularly emotionally challenging, such as treating suicide and self-harm [23]. During the consultation, it was raised that patients with mental health needs may experience distress whilst waiting in the same environment as those with physical health

injuries which warrants further exploration into how to ensure a healing environment for all. Professionals in our study expressed concerns about unintentionally worsening patient outcomes through their language and interactions. This aligns with existing research from patients' perspectives that indicates that stigmatising or uninformed interactions with healthcare providers can exacerbate mental health symptoms, diminish trust, and create barriers to seeking care [43,44]. Based on our findings, and in line with recommendations from mental health patients and nursing students [9,45], we suggest training on non-stigmatizing, non-triggering and empowering communication strategies should be incorporated into professional curricula. Training should also address the underlying causes of mental ill health beyond the biomedical model and stigmatising stereotypes, emphasizing that many of the behaviours perceived as challenging are often trauma responses or coping mechanisms rather than deliberate misconduct or attention-seeking [46]. Enhancing physical health professionals' understanding could foster more empathetic and effective care, reducing the fear of misinterpretation and inappropriate responses.

Our findings indicate that physical health professionals often found managing mental health conditions distressing, with many expressing emotional exhaustion when handling complex mental health needs. Our analysis highlights the importance of professionals having emotional support, as it can be difficult to process how best to support a patient who resists care. Acknowledging that such responses are often symptoms of the illness, rather than personal affronts, may help create a more supportive and compassionate care environment for both staff and patients. Our analysis supports recommendations from the Royal College of Paediatrics and Child Health [25] for more frequent multi-disciplinary meetings along with psychiatric liaison to ensure that physical health professionals are best supported to meet patient's mental health needs. Our findings highlight the potential benefits of cross-disciplinary learning, whereby routine practice from mental health professionals can be adopted by physical health professionals. Our analysis underscored the importance of regular individual clinical supervision and group debriefing for physical health professionals, not only as a response to specific incidents but as an ongoing practice to foster reflection on the emotional demands of their care. Emotional debriefing is a routine procedure amongst mental health professionals, shown to support staff well-being and ultimately contribute to higher-quality patient care [47]. Research on ICU and residential care professionals highlights the effectiveness of emotional debriefing, showing that readily available mental health support enhances staff's ability to provide compassionate care in emotionally demanding environments [48,49]. However, the time required to implement such practices consistently will need to be carefully factored into service planning and staff workloads.

Our analysis supports calls for new collaborative models that address the interconnected needs of young people [50,25]. By emphasizing joined-up working, our study reinforces the importance of integrating mental health and physical health care to provide more comprehensive support for young patients [15]. In line with previous research, our analysis identified technology as a barrier to effective integration, as separate computer systems hindered seamless communication between mental and physical health professionals [51]. In addition, our findings emphasised the need for adequate funding of paediatric care. This analysis identified structural pressures of limited bed availability and short staffing as key challenges for both physical and mental health professionals to best support their patients' health needs. Previous literature has highlighted how chronic understaffing not only increases workload but also limits opportunities for essential training and debriefing sessions, further impacting staff well-being and patient care [52,53]. To enhance collaboration and care coordination, we recommend increased financial and structural support, including the implementation of unified technological platforms that enable real-time information sharing across multidisciplinary teams.

## Limitations

While this study provides valuable insights, it is important to acknowledge its limitations, particularly the nature of our sample. Firstly, it was a self-selecting sample, which may have resulted in the participation of individuals with a specific interest or experience in the topic. This could introduce a degree of response bias, as those who chose to engage in the interviews may differ from those who did not. Secondly, our entire sample identified as female. As many of the teams were

already majority women, this sample could be considered representative of the demographics of these two NHS Trusts. However, as the Kings Fund reports that 76% of the NHS workforce identify as women, the lack of representation from men in this present study cautions against the extrapolation of results to other NHS Trusts, particularly where male representation is higher. Perceptions of safety, experiences of workplace risks, and communication styles have been shown to differ between men and women in various healthcare settings [54], thus our analysis highlighting concerns around staff safety and potential risks in the wards may reflect a gender bias. Future research with a more balanced sample, including a higher proportion of male staff, is needed to explore potential gender-specific differences to gain a more representative account of the opportunities and challenges faced by healthcare professionals supporting mental health needs of children/young people in paediatric wards.

A methodological limitation was the allowance of some interviews to be held in a group setting with colleagues. As explained earlier, this was a pragmatic consideration due to limited time in the working day for staff to participate. We acknowledge that group discussions may limit the voices of those more reserved or hesitant to speak up in front of others and increase the risk of social desirability bias distorting findings, as participants may be inclined to provide responses that align with perceived social or institutional norms rather than their genuine experiences or opinions. However, we do not view this as a significant detriment to the quality or depth of our study data as we feel the diversity of perspectives captured, from a range of health professionals across different roles and settings, helps to strengthen the robustness and validity of our findings.

## Conclusion

This study underscores the significant challenges faced by physical health professionals in caring for young people with both physical and mental health needs. While physical health staff are primarily focused on stabilization and ensuring patient safety, many expressed feeling inadequately prepared to address the complex mental health concerns that frequently present in paediatric settings. The lack of mental health training and resources leaves staff feeling deskilled, leading to potential gaps in care, missed opportunities to develop therapeutic relationships and heightened emotional strain for the professional. These findings suggest the need for integrating targeted mental health education into healthcare training programs, covering areas such as self-harm, eating disorders, suicide risk, and communication skills. These would help better equip professionals with the confidence and competencies required to manage mental health-related situations. The benefit of improved collaboration between physical and mental health services is clear. The positive impact of working alongside psychiatric teams, as reported by participants, further supports calls for integrated care models. Such an approach could bridge existing gaps in service provision, ensuring that young people with complex needs receive more comprehensive, holistic care. Ultimately, a well-coordinated, multidisciplinary system is crucial; not only to strengthen the working environment for healthcare staff but also to ensure young people receive high-quality, comprehensive care that meets both their physical and mental health needs.

## Supporting information

**S1 File. Anonymised interview guide.**
(DOCX)

## Acknowledgments

Thank you to all the participants who took part and all the professionals involved in the consultation sessions which helped to refine our findings. We want to thank Addenbrookes Charity Trust for funding the study. Thank you to Christi Deaton and Amanda Small for supporting the grant application

## Author contributions

**Conceptualization:** Jessica Folwell, Dihini Pilimatalawwe, Tessa Morgan.

**Data curation:** Tessa Morgan.

**Formal analysis:** Jessica Folwell, Dihini Pilimatalawwe, Julia Mannes, Isabella Morse, Tessa Morgan.

**Funding acquisition:** Sara O'Curry, Cathy Walsh, Robbie Duschinsky.

**Investigation:** Jessica Folwell, Tessa Morgan.

**Methodology:** Sara O'Curry, Cathy Walsh.

**Project administration:** Robbie Duschinsky, Tessa Morgan.

**Resources:** Tessa Morgan.

**Supervision:** Tessa Morgan.

**Writing – original draft:** Jessica Folwell, Dihini Pilimatalawwe, Julia Mannes.

**Writing – review & editing:** Jessica Folwell, Dihini Pilimatalawwe, Julia Mannes, Sara O'Curry, Cathy Walsh, Jenny Gibbs, Debbie Critoph, Isabella Morse, Robbie Duschinsky, Tessa Morgan.

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
