## [Decision Letter · Decision Letter 0]

14 Nov 2025

PMEN-D-25-00490

Physical health professional's perspectives of providing mental health care for young people: a qualitative interview study

PLOS Mental Health

Dear Dr. Morgan,

Thank you for submitting your manuscript to PLOS Mental Health. After careful consideration, we feel that it has merit but does not fully meet PLOS Mental Health’s publication criteria as it currently stands. Therefore, we invite you to submit a revised version of the manuscript that addresses the points raised during the review process.

We look forward to receiving your revised manuscript.

Kind regards,

Lambert Zixin Li, Ph.D.

Academic Editor

PLOS Mental Health

Journal Requirements:

Additional Editor Comments (if provided):

Dear Authors,

Thank you for submitting your manuscript to PLOS Mental Health. The reviewers have provided constructive feedback and identified areas that require further clarification and improvement. Based on their comments and my own assessment, I invite you to revise and resubmit your manuscript for further consideration.

Please address all reviewer comments carefully in both the revised manuscript and a detailed response letter. We look forward to receiving your revised manuscript.

Sincerely,

Lambert Zixin Li, PhD

Reviewers' comments:

Reviewer's Responses to Questions

**Comments to the Author**

1. Does this manuscript meet PLOS Mental Health’s publication criteria?

Reviewer #1: Partly

Reviewer #2: Yes

2. Has the statistical analysis been performed appropriately and rigorously?

Reviewer #1: N/A

Reviewer #2: N/A

3. Have the authors made all data underlying the findings in their manuscript fully available (please refer to the Data Availability Statement at the start of the manuscript PDF file)?

Reviewer #1: No

Reviewer #2: Yes

4. Is the manuscript presented in an intelligible fashion and written in standard English?

Reviewer #1: Yes

Reviewer #2: Yes

Reviewer #1: the authors explored an important issues, but the manuscript has the following issues that need to be addressed.

Sample is small (15 only). The authors should say this clearly and why it is enough for their question.

• Participants look too similar (mostly one gender/ narrow roles ). The authors add more variety, or admit this limit properly and explore how this can biase their conclusion

• They used group and one-to-one interviews together; explain what each format added and if answers differ or not. And how they handled themes or response that did not match the majority view

• Some interviews are short; the authors justify how enough depth still came, with typical time given.

• Ethics and consent for re-use must be stated plainly; confirm that approval cover this secondary analysis.

• The authors tell who did interviews/coding, their background, and how bias was reduce.

• Make the analysis steps very clear in 3–4 simple step; add a tiny theme map also if possible

Put more direct quotes to support each theme; add a small table linking theme → quote.

• Use person-first, simple words and define local terms (like “specialing”) for outside readers.

• Do not over-generalise; keep claims to these two trusts only.

• Share de-identified material (interview guide, codebook, quote bank) and say how transcript can be access with request.

• Clean up title, abstract and references; keep consistent British spelling.

Reviewer #2: This is a thoughtful, well-written study of physical health professionals’ (clinicians et al.) perspectives on their experiences taking care of young persons’ mental health (MH) needs in non-MH settings – emergency rooms and pediatric inpatient units. The three main themes emerging from interviews of 15 physical health professionals (3 pediatricians, 5 nurses and health care assistants, 2 managers, 5 allied health care professionals – all women) were feeling out of their depth, their facilities’ being a MH waiting room, and their professional responsibility to keep their MH patients safe in non-MH settings. These themes ring true; they are equally applicable to the circumstances in our county hospital in the United States, where extended ER and general pediatric unit stays are used to compensate for severe lack of pediatric MH facilities.

That this study was commissioned in advance of the building of a new Children’s Hospital and, one hopes, will be used to inform the design of the hospital (especially adequacy of MH spaces in the ER and main wards), is commendable.

All this being said, however, I feel I have been served only half a loaf – where are the corresponding interview analyses and quotes from the 16 MH professionals also interviewed? Given the clear, informative exposition provided thus far for the 15 physical health professionals, I believe similar data about the 16 MH professionals are essential to complement the current presentation. It is likely these 16 MH professionals had comments about the physical health professionals’ being highly anxious about certain MH patients, demanding the MH clinicians do something right away, complaining of the long delays between their being consulted and their appearance at the bedside, etc. (This from our own hospital experiences.). A telling point is that the physical health clinicians believed they had to correct their patients’ physical issues before a MH clinician would provide a consult (lines 314-316). So, I strongly urge the authors to complete this report by providing parallel information on the attitudes and main themes of the 16 MH professionals. This will provide a considerably more powerful and influential report, especially locally for use in the new hospital design, as well as around the world (e.g., in our advocating for increased MH services for young people in our area of the U.S.).

Finally, a few small comments:

Title: …professionals’…. (plural)

Line 84: Physical healthcare professionals are increasingly…

Line 118: …which they are struggling to find… Who are “they”? Could end sentence with …mental health needs.

Line 136: …data were shared. (Plural.)

Line 159: …was reflected (Table 1). (Diversity is singular. Add Table 1 here and move up Table after this paragraph.)

Table 1: All participants were women. Possibility of sex bias; e.g., heightened concern about ensuring the safety of aggressive male adolescents and their own safety?

Line 193: …were complete,

Lines 351-359, 372-375, 383-384: Very important points.

Lines 408-410: A tease; as I strongly urged above, please also present the MH professionals’ findings in this report.

Lines 421-424: This isn’t a limitation; it’s one of the purposes of the report.

References: Please be consistent in the citations; some article titles are lower case, others are in caps. Please space after all the initial numbers (e.g., ref’s 10-13). If DOIs are available, it’s a good idea to include them for ease of searching.

**Do you want your identity to be public for this peer review?** For information about this choice, including consent withdrawal, please see our Privacy Policy

Reviewer #1: **Yes:** Abid rizvi

Reviewer #2: **Yes:** Robert T. Rubin, MD, PhD

---

## [Decision Letter · Decision Letter 1]

5 Jan 2026

Physical and mental health professional's perspectives of providing mental health care for young people: a qualitative interview study

PMEN-D-25-00490R1

Dear Assistant Professor Morgan,

We are pleased to inform you that your manuscript 'Physical and mental health professional's perspectives of providing mental health care for young people: a qualitative interview study' has been provisionally accepted for publication in PLOS Mental Health.

Best regards,

Lambert Zixin Li, Ph.D.

Academic Editor

PLOS Mental Health

Dear Authors,

Thank you for your revision. During the editing process, please address the reviewer’s comments regarding grammatical errors. Otherwise, I am pleased to accept the manuscript for publication.

Sincerely,

Lambert Zixin Li, PhD

Reviewer Comments (if any, and for reference):

Reviewer's Responses to Questions

**Comments to the Author**

Reviewer #2: (No Response)

publication criteria?

Reviewer #2: Yes

3. Has the statistical analysis been performed appropriately and rigorously?

Reviewer #2: N/A

4. Have the authors made all data underlying the findings in their manuscript fully available (please refer to the Data Availability Statement at the start of the manuscript PDF file)?

Reviewer #2: Yes

5. Is the manuscript presented in an intelligible fashion and written in standard English?

Reviewer #2: Yes

Reviewer #2: The authors now have included the interview data of the 16 MH professionals, which I indicated in my review of the original MS are essential to complement the current presentation. This overcomes my major concern with the original version. As well, the authors have attended to all but one of my specific concerns:

The remaining issue is that the MS title should read as …professionals’…, not professional’s (apostrophe should be after the s, not before). The scientific world will see your title in all the citation indexes, and you want it to be correct…

I appreciate the authors’ attentiveness to my critique of the original version.

**Do you want your identity to be public for this peer review?** For information about this choice, including consent withdrawal, please see our Privacy Policy

Reviewer #2: **Yes:** Robert T. Rubin, MD, PhD
